# Design of Outer Membrane Vesicles as Cancer Vaccines: A New Toolkit for Cancer Therapy

**DOI:** 10.3390/cancers11091314

**Published:** 2019-09-06

**Authors:** Yingxuan Zhang, Zheyan Fang, Ruizhen Li, Xiaotian Huang, Qiong Liu

**Affiliations:** 1Department of Medical Microbiology, School of Medicine, Nanchang University, Nanchang 330006, China; 2Key Laboratory of Tumor Pathogenesis and Molecular Pathology, School of Medicine, Nanchang University, Nanchang 330006, China

**Keywords:** cancer vaccines, outer membrane vesicles, antigen carrier, therapy

## Abstract

Cancer vaccines have been extensively studied in recent years and have contributed to exceptional achievements in cancer treatment. They are some of the most newly developed vaccines, although only two are currently approved for use, Provenge and Talimogene laherparepvec (T-VEC). Despite the approval of these two vaccines, most vaccines have been terminated at the clinical trial stage, which indicates that although they are effective in theory, concerns still exist, including low antigenicity of targeting antigens and tumor heterogeneity. In recent years, with new understanding of the biological function and vaccine potential of outer membrane vesicles (OMVs), their potential application in cancer vaccine design deserves our attention. Therefore, this review focuses on the mechanisms, advantages, and prospects of OMVs as antigen-carrier vaccines in cancer vaccine development. We believe that OMV-based vaccines present a safe and effective cancer therapeutic option with broad application prospects.

## 1. Introduction

Cancer is the leading cause of death in economically developed countries and the second leading cause of death in developing countries [1]. Global cancer statistics estimated 18.1 million new cancer cases and 9.6 million cancer deaths in 2018 [2]. For more than 100 years, the hallmark of medical treatment for cancer has been intravenous cytotoxic chemotherapy. New and advanced approaches to cancer research have led to the development of a variety of cancer vaccines in quick succession, and numerous vaccine strategies are currently being evaluated, both pre-clinically and clinically [3,4]. Although traditional cytotoxic chemotherapy remains a treatment backbone for many malignancies, vaccines are now a component of treatment for many types of cancer, including breast, colorectal, lung, and pancreatic cancers; lymphoma; leukemia; and multiple myeloma in preclinical studies, and some are even undergoing clinical trials [4,5,6,7]. However, owing to the characteristics of cancer cells, there are many difficulties in the development of cancer vaccines. The only cancer vaccines currently in routine clinical use are the prostate cancer vaccine Sipuleucel-T (Provenge) and Talimogene laherparepvec (T-VEC) used for the treatment of advanced melanoma; the development of many new vaccines is ongoing with Phase II/III clinical trials [8,9,10,11]. Cancer cells are different from bacteria and viruses in that they are derived from normal cells and exhibit many similarities to somatic cells. Finding a highly efficient vector that allows cancer antigens to activate the immune system and kill or inhibit cancer cell proliferation at this stage is the key to the development of cancer vaccines.

Among the many carriers evaluated for their presentation of cancer antigens, extracellular vesicles (EVs), produced by spontaneous vacuolation of the bacterial membrane, have attracted increasing attention in the research field. Like mammalian cells, Gram-negative and Gram-positive bacteria release nano-sized membrane vesicles into the extracellular environment [12,13]. In recent decades, research into EVs from Gram-negative bacteria has increased substantially, but there is little to no EV-related research with Gram-positive bacteria, because of the inference caused by the thick cell wall of Gram-positive bacteria [14,15]. EVs from Gram-negative bacteria originate from the outer membrane and are thus usually referred to as outer membrane vesicles (OMVs) [16]. OMVs are lipid-based vesicular nanostructures that contain a variety of porins and can carry heterologous substances to accomplish the adjuvant and delivery functions [17]. Owing to their capacity to flexibly transport delivered antigens to endothelial cells or antigen presenting cells (APCs), OMVs have great potential in vaccine development [18,19,20]. Although a few studies have reported the antitumor effect of EVs from Gram-positive bacteria, their formation mechanism and biological function have not been fully elucidated. Therefore, in this review, we mainly focus on OMVs as the EVs from Gram-negative bacteria and analyze the status of cancer vaccine research and the advantages of OMVs as a carrier, focusing on the requirements for designing OMVs as antigen carriers for cancer treatment.

## 2. Vaccines in Cancer Therapy

### 2.1. Historical Overview

A cancer vaccine either prevents infections with cancer causing viruses, treats existing cancer, or prevents the development of cancer in certain high-risk individuals [3,6,21]. Vaccination against an infectious neoplastic agent can be categorized based on three clinical applications: (1) a prophylactic vaccine to prevent infection or acute disease; (2) a therapeutic vaccine to treat an established infection before a malignancy has been induced; or (3) a therapeutic vaccine to treat the infection after the malignant tumor has developed. Prophylactic vaccines are mainly used in viral infections associated with a high risk of cancer, for example, the human papillomavirus (HPV) vaccine [22]. Prophylactic vaccines therefore seem to be an ideal approach to limit the risk of cancer; however, it is difficult to conduct prophylactic trials as the response to the vaccine cannot be evaluated within a short period of time, when compared with suppression of infection or disease in therapeutic vaccine trials [23]. Advances in targeted therapies have reinvigorated interest in cancer immunotherapy because (1) therapeutic vaccines employed as a monotherapy could deliver a targeted immune-mediated effect in tumors, and (2) the advantage of cancer vaccines in comparison with passive immunotherapy is that T-cell driven responses and associated memory may assist during disease recurrence [24]. Since our immune system is built to target and destroy “non-self,” theoretically, cancer vaccination is the safest, most natural, and effective therapeutic approach against cancer. Owing to their ability of immune regulation and presenting multiple heterologous antigens to host cells, OMVs can be a new option for design as an effective therapeutic vaccine to deliver tumor antigens or small molecule drugs to APCs or even targeted cancer cells through genetic engineering.

To date, a group of cancer vaccine models have been developed, including whole-cell and lysed-cell vaccines, gene-modified tumor vaccines, heat shock proteins, peptide-based vaccines, naked DNA, viral vectors, and ex vivo dendritic cell (DC) vaccines [21,25,26,27]. These vaccine models present a good example for the theory that "prevention is better than cure". Their advancement and application will ensure a decrease in cancer incidence globally. At present, the promotion of cancer vaccines in clinical practice is positive, and in general, acceptance of these vaccines is positive. Issues still to be addressed are design of effective cancer vaccines, lack of familiarity with cancer vaccine models, varied beliefs regarding the risk of acquiring infections, and concerns regarding age specifications for the vaccines [3,28,29]. Our focus is on improving the protective effect of the vaccine stimulating to the host and the delivery efficiency of the vaccine vector. Moreover, we believe that OMVs as a tumor antigen carrier can provide us with a novel strategy to design cancer vaccines.

### 2.2. Status of Clinical Trials

A large number of preclinical and clinical studies involving cancer vaccines with varying levels of success have been described over the last two decades. However, with the exception of Sipleucel T, an ex vivo DC vaccine for prostate cancer, and T-VEC, an injectable modified herpes virus for advanced melanoma, no therapeutic cancer vaccine has yet shown clinical efficacy in phase III randomized trials, mainly because of the type of patients recruited in the various clinical studies and reasons including the trial design, the specific vaccination approach, and host-related factors [30,31,32]. Rindopepimut (CDX-110) is a peptide vaccine that targets epidermal growth factor receptor variant III (EGFRvIII) to cure glioblastoma. It showed clinical benefit and significant efficacy in phase II clinical trials; however, phase III trials were terminated as it was deemed likely the study would fail to meet its primary end point [33,34]. Additionally, this antigen was mixed with granulocyte-macrophage colony-stimulating factor (GM-CSF) and used to vaccinate patients with breast cancer in the exploratory phase I–II trials [35]. 

Unfortunately, the results from clinical settings so far have been disappointing. In a cumulative analysis of several vaccine trials including 936 patients with different types of solid tumors, Klebanoff et al. used response rate as a measure of positive outcome and concluded that only 3.6% of patients had an objective benefit from vaccination [36]. However, disappointing, lessons learned from recent studies associated with the immune-suppressive microenvironment of tumor cells have suggested further strategies for improvement [37,38]. Indeed, the clinical success of checkpoint blockades has underscored the role of peripheral tolerance mechanisms in limiting vaccine responses and highlighted the potential for applying combination therapies [39]. In conclusion, the future of cancer vaccines looks bright; however, there remain important problems to be addressed as follows: (1) there is still a need for sufficient quantity of tumor cell antigens; and (2) lasting and effective immune responses are not induced. Furthermore, the knowledge of immune-editing and immune-suppressive mechanisms that operate in each individual patient with cancer is gaining attention in the efforts to design new cancer vaccines.

## 3. New OMV Applications: Tumor Vaccines

### 3.1. Roles of OMVs as Vaccines

OMVs are naturally released by Gram-negative bacteria; they mainly comprise the outer membrane (OM) and periplasmic components [16]. The OM comprises inner and outer phospholipids (PL) and lipopolysaccharide (LPS), combined with interspersed membrane proteins. The lumen of the vesicle may contain a variety of bacterial compounds from the periplasm or cytoplasm, such as proteins, RNA or DNA, and peptidoglycan, which can each act as or encode antigens. The formation of OMVs involves three steps: (1) breakage of connections between OM and peptidoglycan, (2) accumulation of components in the periplasmic space, and (3) activity of specific signal and effector molecules [40]. These processes of OMV formation determine the critical role played by OMVs in the interaction between pathogen and host [41]. 

Since the discovery that OMVs can induce an immune response, many studies have aimed to examine the potential for immunization with OMV components. Toxins can act as adhesins for OMVs and therefore allow vesicles to enter cells using the receptor-mediated endocytic pathway [41]. Common components of vesicles, such as outer membrane protein A (OmpA), could also contribute to host cell entry; this adhesin fully activated macrophage cytokine production when presented in the OMV membrane rather than as a mixture of purified vesicle components [41]. OMVs have proper size (20–200 nm), which is an important factor for their efficient processing by APCs, and can present a range of surface antigens in a native conformation, enabling their entry into lymph vessels and uptake by APCs [42]. The natural properties of OMVs, such as their immunogenicity, capacity to act as self-adjuvants, and potential for uptake by immune cells also make them attractive for application as vaccines against pathogenic bacteria [40]. To date, the first generation of OMV vaccines, Bexsero (Novartis) has been approved for the market against serogroup B *Neisseria meningitidis* [43]. Other OMV-based vaccines, though not yet in clinical trial stage, are still being developed in preclinical studies. Additionally, a variety of applications in OMV research and development have been proposed with a focus on infectious diseases (pneumonia, meningitis, and whooping cough) and specifically enteric diseases (cholera, salmonellosis, and shigellosis), most of which utilize OMV antigen binding (Table 1) [44,45,46,47,48]. 

Furthermore, OMVs exhibit a number of functions that aid bacteria in the process of infecting host cells [16]. The most remarkable example is the use of vesicles as a delivery system for pathogen-associated molecular patterns (PAMPs) [49]. OMVs laden with PAMPs, and other OM components that can affect the course of infection and associated host responses are then delivered into distant host cells. The immunogenic properties of OMVs lead to protective mucosal and systemic bactericidal antibody responses that have been exploited for vaccine purposes. Several studies have also documented B-cell responses to OMVs from various bacteria, such as *Escherichia coli* (*E. coli*), *Salmonella typhimurium,* and *Acinetobacter baumannii* indicating that OMVs can easily be used as antigen delivery systems to generate effective antibody responses [50,51,52,53]. Indeed, there is a growing interest in the use of OMVs as a delivery system of tumor antigens for the design of cancer vaccines.

**Table 1 cancers-11-01314-t001:** Outer membrane vesicle-based vaccine against infection in preclinical test.

OMV Source	Model Establishment	Animal Model	Route of Administration	Adjuvant Used	Protection	Resulting Immune Response	References
*A. baumannii*	Sepsis	Female ICR Mice (6–8 weeks of age)	Intramuscular	Alum	73% survival (compared with 7% in controls)	Serum IgG	[54]
	Acute pneumonia				Reduced bacterial burden in the lungs, spleen and BALF	BALF IgA and IgG	
*B. burgdorferi*	Infection	New Zealand White Rabbits	Intramuscular	Alum	Reduced bacterial burden in the skins	Serum IgG	[55]
*B. pseudomallei*	Septicemic infection	Female BALB/c Mice (8–10 weeks of age)	Subcutaneous	None	50% survival (compared with 0% in controls)	Serum IgG and T_H_1 cell response	[56]
*B. pertussis*	Infection	Female C57BL/6 Mice (8 weeks of age)	Intraperitoneal	Alum	Reduced bacterial burden in the lungs	Serum IgG, T_H_1, and T_H_17 response	[57]
*E. coli*	Sepsis	Both C57BL/6 and BALB/c Mice (5 weeks of age)	Intraperitoneal	None	80%–100% survival (compared with 20% in controls)	Serum IgG, T_H_1, and T_H_17 response	[58]
*E. coli* expressing streptococcal antigen	Sepsis (group A streptococci)	CD1 female Mice	Intraperitoneal	Alum	100% survival (compared with 20% in controls)	Serum IgG1 and IgG2a	[19]
*E. coli* expressing Apx fusion antigen	Porcine pleuropneumonia	Female BALB/c Mice (4–5 weeks of age)	Subcutaneous	Alum	62.5% and 87.5% survival for *Actinobacillus pleuropneumoniae* infection	Serum IgG T_H_1 and T_H_2 cytokines secretion	[59]
Nontypeable *H. influenzae*	Infection	Female BALB/c Mice	Intranasal	None	Reduced bacterial burden in the nasopharynx	Serum IgA, IgG1, and IgM	[20]
*H. pylori*	Infection	Female BALB/c Mice	Intragastric	Cholera toxin	100% protection (compared with 20% in controls)	Serum IgG	[60]
*K. pneumoniae*	Sepsis	Female C57BL/6 Mice (6–7 weeks of age)	Intraperitoneal	None	80%–100% survival (compared with control groups)	Serum IgG and the secretion of key cytokines of T_H_1 cells	[61]
*N. meningitides**	Meningococcal infection	Human use	Intramuscular	Alum	Provided broad-based protection	Not mentioned	[62]
*N. meningitides* with inactivated RSV	Infection (RSV)	Female BALB/c Mice (5–8 weeks of age)	Intranasal	OMVs	100% protection as measured by viral load	IgA plasma cells in NALT, IgA, IgG1, IgG2a, and IgG2b in lung and serum	[63]
*P. gingivalis*	Infection	Female BALB/c Mice (6–8 weeks of age)	Intranasal	Poly (I:C)	Not mentioned	Serum IgG (including IgG1 and IgG2a), salivary S-IgA, and cytokine secretion	[64]
*S. enteritidis*	Foodborne infections	Female BALB/c Mice (6–8 weeks of age)	Intranasal or intraperitoneal	None	83.3%–91% survival (compared with 0% in controls)	Serum IgG and secretory IgA	[65]
*S. typhimurium*	Infection	Female C3H/HeJ and C3H/HeN Mice (6–8 weeks of age)	Intraperitoneal	None	Reduced bacterial burden in spleen, liver, MLNs, and Peyer’s patches	Serum IgG and IFNγ-producing CD4^+^ T cells	[66]
*S. flexneri*	Infection (lethal dose)	Female BALB/c Mice (9 weeks of age)	Nasal or oral	Poly-anhydride nanoparticles	80%–100% protection (compared with 0% in controls)	Serum IgG1 and IgG2a, fecal IgA	[67]
*S. boydii*	Infection (lethal dose)	Female BALB/c Mice (6–7 weeks of age)	Oral	None	100% protection (compared with 0% in controls)	Mucosal IgG and IgA, T_H_1 cell response	[68]
*V. cholerae*	Infection (neonates)	BALB/c Mice (5- to 6-day-old pups)	Intragastric or intranasal	None	Significantly reduced colonization of neonates	Serum IgA, IgG1, IgG2a, and IgM	[69]

*A. baumannii, Acinetobacter baumannii; B. pseudomallei, Burkholderia pseudomallei; B. burgdorferi, Borrelia burgdorferi; B. abortus, Brucella abortus; E. coli, Escherichia coli; H. influenzae, Haemophilus influenzae; H. pylori, Helicobacter pylori; K. pneumoniae, Klebsiella pneumoniae; N. meningitidis, Neisseria meningitidis; P. gingivalis, Porphyromonas gingivalis; S. enteritidis, Salmonella enteritidis; S. typhimurium, Salmonella typhimurium; S. flexneri, Shigella flexneri; S. boydii, Shigella boydii; V. cholerae, Vibrio cholerae*; BALF, bronchoalveolar lavage fluid; IFNγ, interferon-γ; MLN, mesenteric lymph node; NALT, nasal-associated lymphoid tissue; TH, T helper; RSV, respiratory syncytial virus. *The only OMV vaccine licensed to date.

### 3.2. Mechanisms of OMVs Design as Cancer Vaccines

OMVs from Gram-negative bacteria are gaining increasing attention as vaccine candidates for their potential use as carriers of heterologous antigens, presenting the potential for highly effective and easy to produce multi-valent vaccines. Since Kesty and Kuehn first demonstrated the incorporation of heterologous expression of OM and periplasmic proteins in bacterial vesicles [18], several studies have used a variety of strategies to construct recombinant OMVs from different bacteria. Some of the most widely used bacteria for constructing functional OMVs are *Salmonella* and *E. coli*. *Salmonella* have been utilized to produce OMVs, which contain pneumococcal surface protein A (PspA) to protect against *Streptococcus pneumoniae* infection; whereas *E. coli* OMVs containing recombinant HtrA were used in chlamydia vaccine development, and both these strategies were directing target antigens to the lumen of OMVs [70,71]. However, OMVs delivering tumor antigen as cancer vaccines are still in the infancy stage. We aim to explore the preliminary mechanism of OMVs as cancer vaccines for anti-tumor treatment so that more researchers can realize the potential of OMVs as novel cancer vaccines.

The mechanisms involved in the design of recombinant OMVs and underlying the functions of OMVs recombinant with heterologous antigens and pathways through which they gain entry in host cells in vitro have been clarified (Figure 1). The mode of OMV transport to the site of the desired immune response has an important impact on potency. Generally, vaccines are injected subcutaneously or intramuscularly; therefore, transport of antigens through the lymphatic system from the peripheral tissues to the lymphoid organs must be considered in vaccine design. When fluids and serum components circulate between blood capillaries and the interstitial space, peripheral immune cells and antigens or pathogens can enter the lymph vessels. Initial lymph vessels are 10–60 μm in diameter, whereas larger lymphatic vessels can be up to 2 mm in diameter. Bacteria must be carried into the lymphatic system by specialized cells, such as DCs, which can squeeze through openings between overlapping endothelial cells, whereas OMVs, with the size of 20–200 nm, have an intrinsic advantage in transport efficiency [42]. Furthermore, delivery of an OMV-associated antigen in a sulfatase-dependent manner was recently reported, where OMVs were found to traverse the gut mucosal barrier, accessing the gut epithelial cells and underlying intestinal macrophages in a sulfatase-dependent manner, and thus, triggering intestinal inflammation (Figure 1A) [72]. 

Furthermore, the uptake mechanism of OMVs should be identified and internalization of OMVs can be achieved by five different pathways: clathrin-, caveolin-, and lipid raft-mediated endocytosis; membrane fusion pathways; and micropinocytosis (Figure 1B) [49]. Clathrin-mediated endocytosis utilizes dynamin for budding off, and endosomal trafficking routes for vesicle entry; its cargo can then either be returned to the cell surface or targeted to lysosomes for degradation [73]. Based on our understanding of virulence factor transport, it is reasonable to infer that OMVs can utilize toxin-receptor interactions to facilitate their cargo delivery via clathrin dependent endocytosis. The size limit for clathrin-mediated endocytosis is 120 nm (diameter), and therefore, several studies have identified this process as a route for OMV entry, via receptor-ligand binding [74,75]. Since *Helicobacter pylori* OMVs are known to transport vacuolating toxin VacA, an important cytotoxic virulence factor during infection, they have been shown to facilitate their cargo delivery via clathrin-mediated endocytosis [76]. Caveolin-mediated endocytosis occurs via invagination (80 nm) of membranes rich in cholesterol, sphingolipids, and caveolin [77]. Sharpe and colleagues identified this process in nontypeable *Haemophilus influenzae* and showed that OMVs colocalized with the endocytosis protein caveolin, indicating that internalization is mediated by caveolae, which are cholesterol-rich lipid raft domains [78]. Despite sluggish internalization speed (five-times slower than clathrin-mediated), it leads in the efficient delivery of cargo to the cytosol [73]. Lipid rafts are domains of the plasma membrane that are also enriched in sphingolipids and cholesterol [77]. It is hypothesized that clustering of cholesterol-rich regions causes curvature in the membrane, driving movement of molecules into the host cell via invagination (90 nm). Membrane fusion is another mechanism utilized by OMV for entry into host cells; this does not entirely depend on active, energetic processes and the exact mechanisms involved are still under investigation [49,79]. Further, macropinocytosis is characterized by the formation of large (over 200 nm in diameter), actin-driven, ruffled protrusions from the cell membrane [80]. However, macropinocytosis is generally not a cargo-induced process; it allows internalization of endocytic vesicles up to 1 μm in diameter; thus, entry via this route is thought not to be a deliberate OMV-driven event [81]. Therefore, the size of the OMV population is relevant when studying endocytic routes, as macropinocytosis allows internalization of vesicles of 1 μm diameter, while clathrin-dependent endocytosis allows internalization of particles with a maximum diameter of 120 nm. Different isolation methods can introduce a bias towards particular sizes of OMVs, and therefore the purification method is also one of the most important considerations in the design of OMV vaccines.

Through the various mechanisms of entry into host cells described above, OMVs not only interact with mucosal epithelial cells, resulting in the production of cytokines and chemokines and the generation of a pro-inflammatory response, but also gain access to the submucosa beyond the host’s epithelial cell barrier, and directly interact with various immune cell populations, including neutrophils, macrophages, and DCs. The detailed mechanisms through which OMVs induce inflammation and modulate the host immune system have been reviewed [17]. We restrict our discussion to which OMVs modulate presentation of heterologous antigens targeting cancer cells or other immune cells to facilitate the induction of innate and adaptive immunity and thus to achieve the goal of eradicating tumor cells. Although there are relatively few studies directly investigating the anti-tumor immune response induced by OMVs delivering immunogenic antigens at present, we can reveal the mechanisms of OMV-dependent antitumor immune responses by referring to the interaction between OMVs and the host immune system (Figure 1C). OMVs contain numerous PAMPs, including DNA, RNA, lipoproteins, LPS, and peptidoglycan [49]. These PAMPs of OMVs enable them to engage with host pattern recognition receptors (PRRs) such as Toll-like receptor 4 (TLR4) to initiate pro-inflammatory signaling cascades that lead to the production of cytokines and chemokines [82]. Therefore, OMV-delivered antigens as adjuvants could stimulate APCs through the up-regulated expression of receptors and co-stimulatory molecules, and thus, enhanced T helper cells production and fully amplified cellular and humoral immune responses [83,84]. Moreover, OMVs that have entered host cells are detected by intracellular host pattern recognition receptors, like nucleotide-binding oligomerization domain-containing protein 1 (NOD1), which seems to be the key intracellular host pattern recognition receptor that is responsible for the establishment of OMV-dependent immune responses. OMVs can mediate inflammatory signaling via the NOD1 receptor, ultimately resulting in the recruitment and activation of DCs to facilitate the development of T cell immunity [85,86]. Furthermore, OMV delivered antigens are presented by APCs to CD4+ T cells, which leads to the generation of antigen-specific B cell responses [66,87]. Taken together, OMVs as the adjuvant or delivery vector can be easily phagocytized and processed by APCs, thereby promoting adaptive immune responses including cytotoxic T-cell lymphocyte responses that are crucial for tumor and metastasis eradication.

### 3.3. Advantages of OMVs as Cancer Vaccines

Cancer vaccines commonly include three essential components: tumor special antigen or tumor associated antigen, potent adjuvants, and a delivery system. Loss of any of these components can lead to an ineffective immune response or even drive antigen loss and immune evasion [88]. The appropriate selection of tumor antigens has been reviewed [29]. Here, we restrict our discussion to OMVs as the suitable adjuvants and efficient delivery vectors of tumor antigens. Through the suitable design of an OMV, it can be more efficient in delivering tumor antigens or molecule drugs to fulfill antitumor treatment. OMVs can easily be decorated with foreign antigens or epitopes via different synthetic biology approaches, especially by utilizing protein antigen-coding RNA and siRNA interfering gene expression. Furthermore, OMVs are capable of carrying more than one cancer-specific epitope, corresponding to the diversity of tumor antigens (Figure 2). The formation of OMVs involves a sorting mechanism, where proteins are screened and selectively localized to the periplasm or outer membrane. Utilizing this mechanism, different tumor antigens can be delivered to the lumen or surface of vesicles by fusing their coding sequences to those proteins which act as leader peptides for delivery. Some tumor antigens can be located on the surface of OMVs to be better recognized by APCs and thus, induce immune response to kill tumor cells. However, in order to reach deeper tumor tissues to fulfill function, other tumor antigens and small molecule drugs need to be expressed inside the OMVs, which can be hydrophobically encapsulated by lipid bilayers to protect the targeted antigens from degradation of exogenous proteases and early recognition of the immune system [89,90]. Nevertheless, several studies have demonstrated that various antigens delivered to the lumen of OMVs can elicit an effective antibody response and significant protection [19,71].

Initially, the strategy of displaying heterologous antigens into OMVs was to transport the target antigens to the periplasm space by fusing to the N-terminal β-lactamase signal sequence, which was then encapsulated in lumen during the formation of OMVs [71]. Furthermore, Qi Chen et al., tested SlyB as a carrier to direct proteins to the interior of OMVs; cohesin domains were inserted between the Z-domain and INP and functionalized with a dockerin-tagged GFP for cancer cell imaging, indicating the potential role for SlyB and INP as leader peptides to carry antigens [91]. Another delivery system involves fusing antigens with specific proteins such as ClyA, AIDA-I, and OspA which act as leader sequences for antigen delivery [92,93,94]. Currently the most studied leader peptide directing heterologous antigens to the surface of OMVs is ClyA [95,96,97]; genetic fusion between recombinant polypeptides and the C-terminus of ClyA results in a functional display of recombinant protein on the surface of *E. coli* and their derived OMVs [96]. Moreover, several heterologous antigens have also been successfully exported to the surface of OMVs when fused to the β-barrel forming auto transporter AIDA or borrelial lipoprotein OspA [19,94]. In addition to using protein fusion to display antigens, many studies demonstrated that a hemoglobin protease (Hbp) autotransporter platform, originally developed by Jong et al., could also be used to display heterologous antigens on the surface of OMVs [98,99,100]. Taken together, the design of the location of tumor antigen is a complex process, which needs more research to elucidate and verify; especially, various antigen delivery strategies should pave the way for the genetic design of OMVs as cancer vaccines. 

A second factor contributing to OMV potency is their inherent adjuvant properties. Owing to the presence of PAMPs in OMVs, TLR-mediated recognition can occur, thereby driving the inflammatory response in conjunction with complement system activation [101]. PAMPs can also elicit potent Th1-skewed immune responses without the need to add additional adjuvants or delivery systems [83,84]. 

A third aspect is the integrated delivery system. It is well known that tumor cells are ubiquitous, making the drug diffusion process difficult to control, which may lead to multiple-drug resistance [102]. However, two characteristics of OMVs can solve this problem. OMV size is on a nanometer scale (<200 nm), which means that they can extravasate into the tumor tissues via the leaky vessels using the enhanced permeability and retention effect; drugs can then be released into the vicinity of the tumor cells [103,104]. OMVs are stable and rigid and are therefore not susceptible to drug leakage or degradation in the circulation that occurs commonly with polymeric or liposome-based carriers. OMVs also accumulate selectively in tumor tissue when administered systemically. After arriving at the destination, OMVs can kill tumor cells through direct and indirect effects. To deliver siRNA and other proteins or peptides that are toxic to tumor cells, OMVs can recognize and bind to target cells through ligand-receptor interactions. This occurs via components uniquely expressed on the cell surface [102], such as anti-HER_2_ affibody and HER_2_ (overexpressed in the tumor cells) [105]. In this context, bound carriers are internalized before the drug is released inside the cell. While OMVs can enter APCs via receptor-mediated entrance, transport of materials into cells through membrane fusion has also been reported [101]. More importantly, antigen and immune stimulators that enhance the intensity of immune responses and modulate the direction of the response could be delivered to the same APCs simultaneously, facilitating the production of a robust and effective antigen-specific immune response [101].

In conclusion, OMVs have their distinct immunological and structural features, including their nanometer-scale vesicle structure, self-adjuvant properties, ability to be genetically modified, capacity to present large exogenous proteins, and ability to carry immune stimulators. All these features make the OMV an ideal antigen carrier. Currently, many studies are underway to test the applicability of OMVs as vectors in cancer vaccines.

### 3.4. Bacterial OMVs as Cancer Vaccines

Over a century ago, William Coley developed the first bacteria-based cancer treatment by injecting killed bacteria directly into a tumor after having observed regression of the tumor subsequent to bacterial injection. This suggests that live-attenuated bacteria can potentially function as cancer antigen carriers through genetic modification [106]. In addition, the concept of exploiting bacteria as biological tumor vaccine vectors has existed for some time, and the emergence of the anti-tumor strategy of using live bacteria provided the theoretical basis for the development of OMV as a cancer vaccine by shedding from the bacterial outer membrane. Here, we summarize multiple live bacterial vectors used in preclinical research and clinical trials for anti-cancer therapy to better use these live bacterial vectors for the development of OMV-based cancer vaccines (Table 2). Use of bacteria as vectors for cancer vaccines has included intracellular *Salmonella*, *Listeria*, *Pseudomonas aeruginosa* (*P. aeruginosa*), and *E. coli*, for which escape from the phagosome requires virulence factors like listeriolysin O and phospholipase C to degrade the phagosomal membrane and migrate to neighboring cells by a direct cell-to-cell transfer mechanism. In addition, *Listeria* and *Salmonella* vectors have been shown to activate innate immunity. This occurs through TLR binding, secretion of inflammatory cytokines and chemokines, upregulation of co-stimulatory molecules, stimulation of antigen specific CD4+ and CD8+ T cells, and suppression of regulatory T cells [107,108]. Thus, these two attenuated bacteria are commonly used to deliver cancer antigens. Several other Gram-positive bacteria have also been used, including *Clostridium* and *Bifidobacterium*, and these organisms can utilize MHC II molecules to deliver antigens [109]. Therefore, these Gram-positive bacteria can also purify their EVs for cancer vaccine development (Table 2) [110,111]. Although many live bacteria have been used as cancer vaccines in preclinical studies, the biggest problem hindering the entry of such vaccines into clinical trials and clinical application is the balance between safety and immunogenicity. Due to its safety and high efficiency, OMVs provide new choices and ideas for the application of these bacterial vectors as cancer therapeutic vaccines.

OMVs stimulate both humoral and cell-mediated immunity in a manner similar to bacteria; however, they are superior to bacteria in their safety profile and ease of production. Moreover, while intracellular bacteria could be used as live delivery vectors, they could also be utilized to purify OMVs genetically engineered for cancer therapy; several researchers have attempted this and achieved positive results. Gujrati et al., established the Affi_HER2_ OMV vaccine by utilizing siRNA to target kinesin spindle protein and select human epidermal growth factor receptor 2 (HER2) as the receptor in tumor cells [105]. They derived OMVs displaying a HER2-specific affibody from a mutant *E. coli* strain and injected them into mice. The results indicate that systemic injection of siRNA-packaged OMVs caused targeted gene silencing and induced highly significant tumor growth regression in an animal model. More importantly, the modified OMVs were well tolerated and showed no evidence of nonspecific side effects [105]. Moreover, Wang et al., selected HPV16 E7 protein as a tumor antigen and utilized the Trx protein to present E7 both on the surface and in the lumen of OMVs [101]. In contrast to siRNA recombinant OMVs, these were taken up rapidly by DCs, significantly stimulating the expression of DC maturation markers CD80, CD86, CD83, and CD40. As a result, numbers of interferon-gamma (IFN-γ)-expressing splenocytes and IFN-γ-expressing CD4+ and CD8+ cells increased, and growth of grafted TC-1 tumors in mice was significantly suppressed [101]. Other authors have also investigated recombinant OMVs, for example, EGFRvIII-OMV, which demonstrates the decoration of OMVs with multiple antigens, further corroborating the protective efficacy of the vaccine [88]. Further, given the wide use of OMVs as non-living complex vaccines or delivery vehicles, Oh Youn Kim and colleagues firstly demonstrated the function of OMVs in treating cancer [112]. This showed remarkable capability of *E. coli* OMVs to effectively induce a long-term antitumor immune response that can fully eradicate the established tumor without notable adverse effects. In particular, this study also showed for the first time that bacterial EVs derived from Gram-positive bacteria *Lactobacillus acidophilus* and *Staphylococcus aureus* observed significant antitumor effects, suggesting the potential of using EVs derived from antitumor live Gram-positive bacteria in future clinical applications [112]. Furthermore, these results revealed the potential mechanism of OMV accumulation in tumor tissue and production of IFN-γ within the tumor microenvironment to activate antitumor response and expanded the application of OMVs as cancer vaccines not only as delivery platforms but also as antitumor agents. 

**Table 2 cancers-11-01314-t002:** Major live bacterial vectors used for cancer treatment and showing the potential of OMV- or extracellular vesicle (EV)-based cancer vaccines.

Vaccine Strain	Gene Mutated or Modified	Descriptions	Therapeutic Agents (Tumor Antigen, Immune molecule, Anti-Tumor Drug)	Prokaryotic/ Eukaryotic Expression	Cancer Indication	Clinical Trials	Therapeutic Efficacy#	References
***Salmonella***								
VNP20009	*msbB*/*Pur*	The deletion of *msbB* modifies the lipid-A structure reducing bacterial ability of TNF-α induction and mutation of *pur* results in bacterial deficiency in adenine synthesis	**Cytokines:** IL-18, LIGHT, CCL21; **Cytotoxic agents:** TRAIL, FasL; **Regulators:** Thrombospondin; **TAA:** CEA-scFv, TGF-α; **Prodrug enzymes:** Cytosine deaminase; **si-RNA:** sox2-specific, IDO-specific	Prokaryotic or eukaryotic expression	Colon, lung, breast, cervical melanoma	Phase I (used for metastatic melanoma)	Safety, and targeting to tumor cells, but no patients experienced objective tumor regression in Phase I clinical trial	[113,114,115,116,117,118,119,120,121,122]
SL3261 SL7207 et. al.	*aro-*	The genes *aroA* and *aroD* are responsible for the biosynthesis of aromatic amino acids	**Cytokines:** IL-2, IL-12, IL-4, IL-18, IFN-γ, GM-CSF; **Cytotoxic agents:** cytolysin A, Noxa, FlaB, apoptin, diphtheria toxin; **Regulators:** 4-1BBL; **TAA:** CD20-specific antibody, HPV16-E7, Survivin, FLK-1; **si-RNA:** Bcl-2-specific	Prokaryotic or eukaryotic expression	Osteosarcoma, melanoma, colon, breast, cervical, gastric, neuroblastoma, lung, prostate	None	Effectively suppressed tumor growth and metastasis in mouse model	[123,124,125,126,127,128,129,130,131,132,133,134,135,136,137,138,139]
SHJ2037	*relA*/*spoT*	Lacking both RelA and SpoT, cells are unable to produce ppGpp, a global regulator involving bacterial adaptation of extreme environment	**Regulator:** L-asparaginase, **TAA:** RGD peptides, TGF-α	Prokaryotic expression	Colon, breast	None	Effectively suppressed various solid tumor growth in mouse model	[140,141]
ST8	*asd*/*gmd*	The gene *gmd* is in the colanic acid gene cluster and encodes GDP-mannose 4,6-dehydratase; *Salmonella asd* mutants will lyse during growth unless exogenous DAP is supplied	**Regulator:** Endostatin	Prokaryotic expression	Colon	None	Successfully suppressed angiogenesis and consequently retards tumor growth	[142]
LH430	*phoP*/*phoQ*	The knock-out of *PhoP* and *PhoQ* that regulate acid phosphatase synthesis significantly reduces bacterial survival in macrophages	**Regulators:** Endostatin; **siRNA:** STAT3-specific	Eukaryotic expression	Hepatoma	None	Stimulated apoptosis and inhibited angiogenesis in tumors	[143]
MvP728	*purD*/*htrA*	The gene *purD* encodes 5′- phosphoribosylglycinamide synthetase involved in purine biosynthesis; htrA encodes heat-shock proteins that are important for virulence of the bacterium	**TAA:** survivin	Eukaryotic expression	Glioblastoma, colon	None	Enhanced effector-memory CTL response and inhibited tumor growth in mouse model	[144]
χ4550	*cya*/*crp*	The two genes *cya* and *crp* encode cAMP (cyclic adenosine monophosphate) synthetase and cAMP receptor protein	**Cytokines:** IL-2, TNF-α	Prokaryotic expression	Melanoma	None	Inhibited tumor growth as well as enhanced host survival	[145]
RE88	*dam*	The gene *dam* encodes DNA, adenine methylase sptP is an effector protein of *Salmonella*	**TAA:** legumain	Eukaryotic expression	Breast	None	Effectively suppressed tumor angiogenesis	[146]
SB824	*sptP*	SptP is an effector protein of Salmonella pathogenicity island 1 (SPI-1), that acts as protein tyrosine phosphatase/GTPase activating proteins	**TAA:** YopE1-138/p60130-477/M45	Prokaryotic expression	Fibrosarcoma	None	Showed complete tumor regression	[147]
***Listeria****								
ADXS11-001 ADXS31-142 ADXS31-164 ADXS-NEO et. al.	*tLLO* (Lysteriolysin)	Lm has the ability to replicate in the cytosol of APCs after escaping from the phagolysosome, which requires the virulence factor listeriolysin O (LLO) protein, and targeted antigen fused to a non-hemolytic LLO	**TAA:** HPV 16 E7, PSA, VEGFR2, HER2, Personal Neo-antigens	Prokaryotic expression	Cervical, oropharyngeal, prostate, colon, lung, breast, HER2^+^ solid tumors	Phase I (used for cervical cancer)	36% survival for 12 months and 11% response rate were observed in patients	[148,149,150]
CRS-100 CRS-207 ADU-623 ADU-214 et. al.	*actA*/*inlB*	Two virulence genes, *actA* and internalin B (*InlB*), and their combined deletion results in 1000-fold attenuation when compared to wildtype	**TAA:** Mesothelin, EGFRvIII, NY-ESO-1, Personal Neo-antigens	Prokaryotic expression	Pancreatic, lung, ovarian, mesothelioma, prostate	Phase I/II (used for pancreatic cancer and mesothelioma	37% of patients survived 15 months or more, and the combination with chemotherapy is more effective	[151,152]
Attenuated *L. monocytogenes*	*dal*/*dat*	In the absence of dal and dat expression, replication of LM can depend only on the availability of exogenous D-alanine. After introduction of the *dal* and *dat* genes from *Bacillus subtilis*, the strain was able to synthesize D-alanine and to replicate to a limited extent that did not cause severe organ injuries	**TAA:** CD24	Eukaryotic expression	Hepatocellular carcinoma	None	Significantly reduced the tumor size in mice and increased their survival from 0% to 48%	[153]
***Clostridium****								
*C. beijerinckii*	Non	*E. coli* nitroreductase known to activate the nontoxic prodrug CB 1954 to a toxic anticancer drug	**Prodrug enzymes:** CB1954	Prokaryotic expression	Breast	None	Lack of toxicity and highly selective growth in tumors	[154]
*C. acetobutylicum*	Non	*C. acetobutylicum* was genetically engineered to express and secrete either mTNF-alpha, IL-2, or the *E. coli* cytosine deaminase	**Cytokines:** TNF-α, IL-2; **Cytotoxic agents:** cytosine deaminase	Prokaryotic expression	Rhabdomyosarcoma	None	Safety and selective colonization pattern	[155]
***Bifidobacterium****								
*B. longum*	Non	*Bifidobacterium* can selectively germinate and grow in the hypoxic regions of solid tumors after intravenous injection	**Cytotoxic agents:** cytosine deaminase, TRAIL; **Prodrug enzymes:** 5-fluorocytosine; **TAA:** Wilms’ tumor 1	Prokaryotic expression	Lung, melanoma; leukemia	None	Selectively proliferated in tumors and significantly suppressed tumor weight and tumor growth	[156,157,158,159]
*B. adolescentis*	Non	A shuttle vector, pBV220 was used for expressing antigens	**Regulator:** Endostatin	Prokaryotic expression	Liver	None	Tumor growth in mice was inhibited by 23.1%	[160]
*B. infantis*	Non	*B. infantis* can selectively localize and proliferate in the hypoxic environment in several types of solid tumors	**TAA:** sFlt-1	Eukaryotic expression	Lung	None	Inhibit the tumor growth and prolong survival time from 41 days to 51 days	[161]
***Pseudomonas aeruginosa***								
CHA-OST et. al.	*exoS*/*exoT*/*aroA*/*lasI*, fusion with EXO-S	An attenuated live bacterial vector using the type III secretion system (TTSS) of *Pseudomonas aeruginosa* to deliver in vivo tumor antigens	**TAA:** TRP-2, gp100, MUC18	Prokaryotic expression	Glioma	None	100% protection in prophylactic antitumor assay and 37.5% protection in therapeutic antitumor assay	[162,163]
***Escherichia coli***								
χ6212	*asd,* SAH was cloned into *E. coli*	*Staphylococcus aureus* α-hemolysin (SAH) is a pore-forming protein that is naturally secreted and kills mammalian cells	**Cytotoxic agents:***Staphylococcus aureus* α-hemolysin (SAH)	Prokaryotic expression	Breast	None	Tumor volume was 59% of induction compared with control group	[164]

TNF-α, tumor necrosis factor-α; IL, interleukin; TAA, tumor-associated antigens; TRAIL, Tumor necrosis factor related apoptosis-inducing ligand; VEGFR2, vascular endothelial growth factor receptor 2; IFN-γ, interferon-γ; GM-CSF, Granulocyte-macrophage colony-stimulating factor; TGF, transforming growth factor; # Unless stated otherwise, the therapeutic efficacy mentioned here referred to the preclinical study in mice. * Sheds or vesicles of Gram-positive bacteria can be spontaneously produced by the exfoliation of cell membranes and are usually named as extracellular vesicles (EVs); thus, Gram-positive bacteria can also be used as targets for the design of vesicles-based cancer vaccines.

## 4. Conclusions and Future Directions 

Currently, OMVs have immense potential to be used as cancer vaccines, but many studies focus only on the insertion of cancer antigens into OMVs and directly use OMVs derived from conventional engineering of *E. coli*. Genetic engineering of OMVs presents enormous potential to artificially modify different intracellular bacteria such as *Salmonella* and *Listeria* to maximize the ability of OMVs to stimulate immune responses, thereby designing an ideal cancer vaccine. In addition, tumor-targeted Gram-positive bacteria such as *Clostridium* and *Bifidobacterium* also spontaneously produced EVs through the exfoliation of cell membranes [110,111]. Furthermore, bacterial EVs derived from Gram-positive bacteria *L. acidophilus* and *S. aureus* showed significant antitumor effects [112]. Thus, Gram-positive bacteria can also be used as targets for the design of vesicle-based cancer vaccines (Table 2).

The challenges that may have to be addressed in OMV cancer vaccine development are as follows: (1) selection of suitable antigens for different types of cancer, (2) enhancing tissue phagocytosis through genetic engineering, and (3) refining the mechanisms of vaccine-induced immune responses and their clinical efficacy [28]. Similarly, the following issues remain to be addressed for OMVs: (1) the high reactogenicity of PAMPs, such as LPS, (2) low expression levels of relevant protective antigens, (3) immuno-dominant antigens that misdirect the immune response, and (4) molecules that are immunosuppressive or otherwise interfering with a protective immune response. Furthermore, among all challenges, immune responses directed to the carrier itself may affect the repeated use of a vaccine based on the same antigen delivery platform. It was reported that pre-existing anti-carrier antibodies may mediate antibody-dependent phagocytosis and promote antigen clearance, thus reducing antigen uptake by APCs and weakening the subsequent immune responses. In addition, undesired responses to OMV components may excessively consume the immune response and thus attenuate specific responses to the target antigen [101]. Therefore, finding a balance between the immune response induced by the OMV itself and the efficiency of antigen delivery is the most important task in designing OMVs as cancer vaccines.

In conclusion, there are still many opportunities and challenges in the development of OMVs, which require more effort and clinical trials. We believe that cancer vaccines based on OMVs can become a safe and effective therapeutic option with prospects for broad application prospects. 

## Figures and Tables

**Figure 1 cancers-11-01314-f001:**
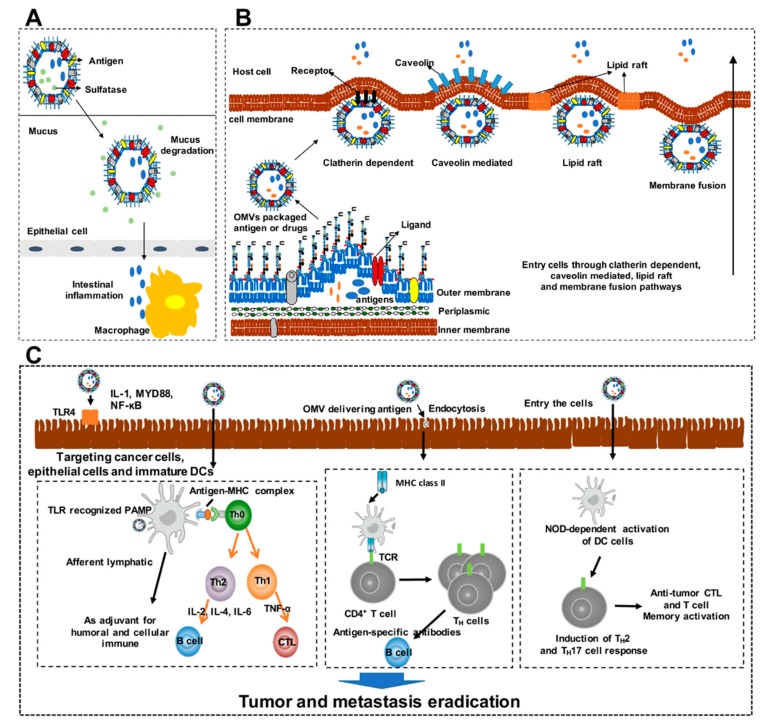
Design of mechanisms underlying the functions and immune modulation of outer membrane vesicles (OMVs) recombinant with heterologous antigens and the pathways through which they gain entry into host cells. (**A**) OMVs can also cross the mucus barrier in the gut and reach the intestinal epithelium, delivering bacterial antigens to the underlying macrophages, triggering intestinal inflammation. (**B**) Mechanisms of OMV entry. Pathogenic Gram-negative bacteria are thought to utilize OMVs to interact with host cells during infection. For example, bacteria can use OMVs to mediate the delivery of virulence factors, such as toxins, into host cells, including immune cells, and OMVs may enter host cells through various endocytic routes including clathrin-dependent, caveolin-mediated, lipid raft, and membrane fusion pathways. The most frequently reported mode of OMV entry into host cells involves lipid rafts as OMVs could fuse with lipid rafts to facilitate their entry into host cells. The pathways of cholesterol-independent and clathrin-mediated endocytosis are independent of lipid rafts. Moreover, OMVs can enter host cells via the mechanism of membrane fusion in a size-dependent manner. (**C**) A model for OMVs targeting cancer cells, epithelial cells, and immature dendritic cells (DCs) to mediate immune responses. OMVs can interact directly with epithelial cells and immune cells or they may interact with pattern recognition receptors, such as Toll-like receptor 4 (TLR4), to induce the production of cytokines and chemokines. OMV adjuvant delivered antigen could be recognized by DCs that led to the recruitment of immune cells and stimulated antigen-presenting cells (APC) through toll like receptor (TLR) recognized pathogen-associated molecular patterns (PAMPs). These process enhanced T helper cells production (including Th1 and Th2), and fully amplified cellular and humoral immunity. Furthermore, OMVs delivering antigens can also traffic into non-immune cells and load onto MHC class II molecules. Activated antigen-presenting cells express MHC class II molecules that interact with the T cell receptor (TCR) on CD4+ T cells to drive antigen-specific T cell responses, resulting in T helper cell proliferation, thereby generating antigen-specific antibodies in various tissues. Following entry into host cells, OMVs are also detected by nucleotide-binding oligomerization domain-containing protein 1 (NOD1). Detection and degradation of intracellular OMVs results in the recruitment and activation of DCs to facilitate the development of T cell immunity (Th2 and Th17).

**Figure 2 cancers-11-01314-f002:**
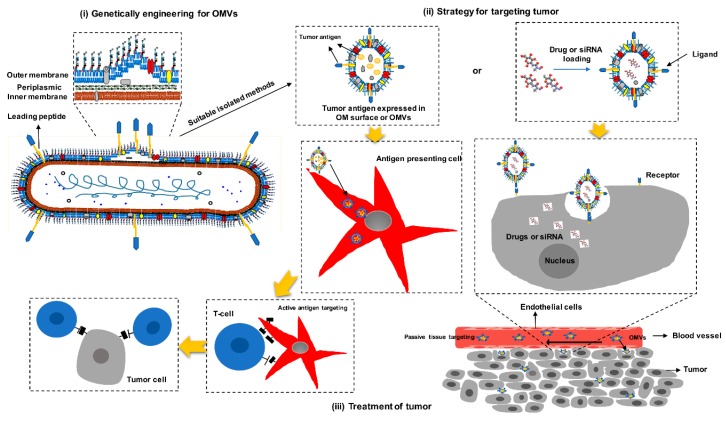
Schematic representation of mechanisms by which OMVs act as carriers to deliver tumor antigens in cancer therapy. Tumor antigens are targeted to the membrane by the leading peptide. OMVs are released from mutant bacterial cells and carry tumor antigens. OMVs travel via blood vessels to arrive at the target tissue. Cells mainly recognize OMVs by ligand-receptor interaction and internalize OMVs by membrane invagination or membrane fusion.

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
