# Peer review of "Design of Outer Membrane Vesicles as Cancer Vaccines: A New Toolkit for Cancer Therapy"

_cancers, 2019, doi:10.3390/cancers11091314_

Round 1

Reviewer 1 Report

OMV-based cancer vaccines is a fascinating area of research and the authors have covered a broad range of information relating to their function and potential use in this paper. Unfortunately, in its current form, it didn’t leave me a clear impression of where the field sits and where it is heading to next. The manuscript would benefit from some re-organisation and refining of information, specifically with regards to: 

1)      It needs to be made clear throughout the paper whether the vaccines discussed are in preclinical or clinical testing or are in clinical use. For example, on page 1 line 30-32 the statement “vaccines are now a component of treatment for many types of cancer” is misleading. There are many vaccines under development but they are not yet part of routine clinical care – as the authors go on to imply when discussing the only 2 cancer vaccines currently in clinical use. There were several points throughout the paper where I wasn’t entirely sure at what stage of testing different vaccines were.

2)      Some of the organisation of sections seemed counterintuitive. There is some back and forth between discussing OMV vaccines and vaccines based on live or attenuated bacteria that gets difficult to follow.  I also think that it would be more useful to have a table detailing what work has been done in preclinical and clinical models with OMVs (as OMVs are the focus of the paper) rather than the lengthy table on bacterial vectors. With the table in its current form I find it difficult to link the information presented to how it relates to OMV-specific vaccines.

3)      Parts of the manuscript would benefit from the addition of more specific information. For example, on page 2 line 82-83 the authors mention a terminated phase III clinical trial but do not mention why this trial was terminated (it would be interesting to know). The authors also mention a “most commonly used commercial OMV vaccine” on page 3 line 107 – most commonly used in what context?  Do you meant that it is the most commonly used for experimental work?  Or for vaccination against infectious disease? Similarly, on page 3 line 128 a first generation OMV vaccine that has been approved for market is mentioned.  What is this vaccine for? Presumably not cancer.  Conversely, there is in some places an excess of information that does not seem entirely pertinent to the topic, e.g. frequent historical references and some repetition of information in sections such as the “Advantages of OMVs as tumor vaccines”.

Minor points:

Page 1 line 24: “Global cancer statistics report that there will be”, this is referring to a 2018 statistic, it should be the past tense “were”.

Page 1 line 44: “capacity to flexibly transport their contents to specific sites” what does this mean?

Page 2 line 7: T-VEC has already been defined in the introduction.

Page 2 line 69: “Cancer vaccine holds a glove in reducing exposure risk” what is meant by this?

Page 2 line 51: “prevents the development of cancer in certain high-risk individuals” please give a reference for this.  You should also provide references for the statements in lines 71-74 of the same page.

Page 3 line 123: “OMVs have a proper size”, what is a ‘proper’ size?

I’d recommend reducing the number of abbreviations in the article – especially where some appear to not be used again.

It would be useful to clarify what type of extracellular vesicles gram positive bacteria produce and how these fit in with OMV vaccines. The vesicles themselves weren’t really mentioned until the conclusions and future directions section (with some mention of gram positive bacteria earlier).

Author Response

Reviewer 1

OMV-based cancer vaccines is a fascinating area of research and the authors have covered a broad range of information relating to their function and potential use in this paper. Unfortunately, in its current form, it didn’t leave me a clear impression of where the field sits and where it is heading to next. The manuscript would benefit from some re-organisation and refining of information, specifically with regards to:

Our response: We appreciate the reviewer’s suggestions, and with the help of reviewers, the quality of our manuscripts has been greatly improved. Our review mainly summarizes the mechanism, advantages and prospects of OMV as a novel cancer vaccine. Although there are few studies on the design of cancer vaccines using OMV at the present stage, it is a new choice for the development of cancer vaccines. We hope that through our review, more researchers will be involved in the development of OMV-based cancer vaccine. We have reorganized most of the content of the full text and tried our best to cover all the questions raised by the reviewer, and we have added new Table 1 and redrawn the new Figure 1 in revised manuscript. Detail revisions have been highlighted in the revised version of the manuscript, and also can refer to the following responses to each question. We hope that our revisions will meet the requirements of the reviewer.

It needs to be made clear throughout the paper whether the vaccines discussed are in preclinical or clinical testing or are in clinical use. For example, on page 1 line 30-32 the statement “vaccines are now a component of treatment for many types of cancer” is misleading. There are many vaccines under development but they are not yet part of routine clinical care – as the authors go on to imply when discussing the only 2 cancer vaccines currently in clinical use. There were several points throughout the paper where I wasn’t entirely sure at what stage of testing different vaccines were.

Our response: We appreciate the reviewer for pointing out this misleading. We also realized that we did not specify in the old manuscript whether the vaccine cases we mentioned were preclinical studies, clinical trials or approved clinical applications. We have revised all issues in the new version of manuscript. And we have made the following modifications in the full text.

We have replaced these sentences in lines 30-32 of old version by “vaccines are now a component of treatment for many types of cancer, including breast, colorectal, lung, and pancreatic cancers; lymphoma; leukemia; and multiple myeloma in preclinical studies, and some are even undergoing clinical trials [4-7]. However, due to the characteristics of cancer cells, there are many difficulties in the development of cancer vaccines. The only cancer vaccines currently in routine clinical use are the prostate cancer vaccine Sipuleucel-T (Provenge), and Talimogene laherparepvec (T-VEC), which is used for the treatment of advanced of melanoma; the development of many new vaccines has ongoing in Phase II/III clinical trials [8-11]” in lines 31-38 of revised manuscript, and we have marked the modification place using red colour. We have replaced the descriptions of lines 129-135 of old version by “To date, the first generation of OMV vaccines, Bexsero (Novartis) has been approved for the market against serogroup B Neisseria meningitides (MenB) [43]. Other OMV-based vaccines, though not yet in clinical trial stage, are still being developed in preclinical studies. Additionally, a variety of applications in OMV Research and Development have been proposed with a focus on infectious diseases (pneumonia, meningitis, and whooping cough) and specifically enteric diseases (cholera, salmonellosis, and shigellosis), most of which utilize OMV antigen binding (Table 1) [44-48].” in lines 139-145 of revised manuscript. And in order to give readers a better understanding of the research prospective of OMV-related vaccines and the stage of these researches, Table 1 is added to illustrate this issue. We added clinical trial summaries of various studies to the new Table 2, and other studies that were not explicitly stated were in preclinical studies. We believe that these revisions will give the reader and reviewers a clear idea of the period in which the vaccine cases we have listed are being studied.

2) Some of the organisation of sections seemed counterintuitive. There is some back and forth between discussing OMV vaccines and vaccines based on live or attenuated bacteria that gets difficult to follow. I also think that it would be more useful to have a table detailing what work has been done in preclinical and clinical models with OMVs (as OMVs are the focus of the paper) rather than the lengthy table on bacterial vectors. With the table in its current form I find it difficult to link the information presented to how it relates to OMV-specific vaccines.

Our response: Yes, we appreciate the reviewer’s suggestion. And we have added the new Table 1 to illustrate the major studies in preclinical and clinical models with OMVs. Meanwhile, we listed live bacterial vectors mainly to illustrate that these bacteria, which can be used as living bacterial vectors, also spontaneously produce OMVs, which means that they have the potential to be the target of developing OMV-based cancer vaccines. Therefore, we keep the table in revised manuscript. Since the research of OMVs as cancer vaccines is still in its infancy, we can not directly link these live vector vaccines with OMV.

And that, we have also reorganized these paragraphs to illustrate the relationship between OMV design and live bacterial vectors. Details are shown below.

Over a century ago, William Coley developed the first bacterial-based cancer treatment by injecting killed bacteria directly into a tumor after having observed regression of the tumor subsequent to bacterial injection. This suggests to us that live-attenuated bacteria can potentially function as cancer antigen carriers through genetic modification [101]. And that, the concept of exploiting bacteria as biological tumor vaccine vectors has existed for some time, and the emerge of anti-tumor strategy of live bacteria provided the theoretical basis for the development of OMV shedding from bacterial outer membrane as cancer vaccine. Here, we summarize multiple live bacterial vectors used in preclinical research and clinical trials for anti-cancer therapy, in order to better use these live bacterial vectors for the development of OMV-based cancer vaccines (Table 2). Use of bacteria as vectors for cancer vaccines has included intracellular Salmonella, Listeria, aeruginosa, and E. coli, for which escape from the phagosome requires virulence factors, such as listeriolysin O and phospholipase C, degrading the phagosomal membrane and migrate to neighboring cells by a direct cell-to-cell transfer mechanism. And that, L. monocytogenes and Salmonella vectors have been shown to activate innate immunity. This occurs through TLR binding, secretion of inflammatory cytokines and chemokines, upregulation of co-stimulatory molecules, stimulation of antigen specific CD4+ and CD8+ T cells, and suppression of regulatory T cells [102,103]. Thus these two attenuated bacteria are commonly used to deliver cancer antigens. Several other Gram-positive bacteria have also been used, including Clostridium, and Bifidobacterium, and these organisms can utilize MHC II molecules to deliver antigens [104]. Therefore, these Gram-positive bacteria can also purify their vesicles for cancer vaccine development, although the vesicles of Gram-positive bacteria are called extracellular vesicles (Table 2) [105,106]. Although many live bacteria have been used as cancer vaccines in preclinical studies, the biggest problem hindering the entry of such vaccines into clinical trials and clinical application is the balance between safety and immunogenicity. Because of its safety and high efficiency, OMVs provide new choices and ideas for the application of these bacterial vectors as cancer therapeutic vaccines” at lines 358-381 of the part “3.4 Bacterial OMVs as cancer vaccines”. At present, OMVs have great potential to be used as cancer vaccines, but many studies only involve the insertion of cancer antigens into OMVs and directly use OMVs derived from conventional engineering of coli. Genetic engineering of OMVs presents great potential to artificially modify different intracellular bacteria such as Salmonella and Listeria, to maximize the ability of OMVs to stimulate immune responses, thereby designing an ideal cancer vaccine.” at lines 419-423 of the part “4. Conclusion and future directions”.

3) Parts of the manuscript would benefit from the addition of more specific information. For example, on page 2 line 82-83 the authors mention a terminated phase III clinical trial but do not mention why this trial was terminated (it would be interesting to know). The authors also mention a “most commonly used commercial OMV vaccine” on page 3 line 107 – most commonly used in what context?  Do you meant that it is the most commonly used for experimental work?  Or for vaccination against infectious disease? Similarly, on page 3 line 128 a first generation OMV vaccine that has been approved for market is mentioned.  What is this vaccine for? Presumably not cancer. Conversely, there is in some places an excess of information that does not seem entirely pertinent to the topic, e.g. frequent historical references and some repetition of information in sections such as the “Advantages of OMVs as tumor vaccines”.

Our response: We thank the reviewer point out these problems. For the first question, we have modified these sentences and replaced these by “However, with the exception of Sipleucel T, an ex vivo DC vaccine for prostate cancer, and T-VEC, an injectable modified herpes virus for advanced melanoma, no therapeutic cancer vaccine has yet shown clinical efficacy in phase III randomized trials, mainly due to the type of patients recruited in the various clinical studies and other reasons including the trial design, the specific vaccination approach, host-related factors [23-25]. Rindopepimut (CDX-110) is a peptide vaccine that targets epidermal growth factor receptor variant III (EGFRvIII) to cure glioblastoma; it showed clinical benefit and significant efficacy in phase II clinical trials; however, phase III trials were terminated, as it was deemed likely the study would fail to meet its primary end point” in lines 34-38 of revised manuscript.

For the second question, the most common use mentioned here is for vaccine development. And considering the coherence of the context and not to be misunderstood by the reader, we decided to delete this sentence in revised manuscript. Further, the Bexsero vaccine have been approved against serogroup B Neisseria meningitides (MenB) infection and we have modified this sentence in lines 139-145 of revised manuscript.

At the prompt of the reviewer, we also realized that there were repetitive problems in our paper. We have reorganized these paragraphs in new version of manuscript. Among them, the genetic engineering of OMV is repeated in parts of 3.2 and 3.3, so we move all contents to the part of 3.3 to describe and we have reorganized these sentences. The details are as follows: “Initially, the strategy of displaying heterologous antigens into OMVs was to transport the target antigens to periplasm space by fusion to the N-terminal β-lactamase signal sequence, which was then encapsulated in lumen during the formation of OMVs [73]. Furthermore, Qi Chen et al. tested SlyB as a carrier to direct proteins to the interior of OMVs; cohesin domains were inserted between the Z-domain and INP and functionalized with a dockerin-tagged GFP for cancer cell imaging, indicating the potential role for SlyB and INP as leader peptides to carry antigens [95]. Another delivery system involves fusing antigens with specific proteins such as ClyA, AIDA-I and OspA, which act as leader sequences for antigen delivery [96-98]. Currently the most studied leader peptide directing heterologous antigens to surface of OMVs is ClyA [99-101]; genetic fusion between recombinant polypeptides and the C-terminus of ClyA results in a functional display of recombinant protein on the surface of E. coli and their derived OMVs [100]. Moreover, several heterologous antigens have also been successfully exported to the surface of OMVs when fused to the β-barrel forming auto transporter AIDA or borrelial lipoprotein OspA [19,98]. In addition to using protein fusion to display antigens, many studies demonstrated that a hemoglobin protease (Hbp) autotransporter platform, originally developed by Jong et al., could also be used to display heterologous antigens on the surface of OMVs [102-104]. Taken together, the design of the location of tumor antigen is a complex process, which needs more research to elucidate and verify, but even then various antigen delivery strategies should pave the way for the genetically design of OMVs as cancer vaccines” in lines 304-322 of revised manuscript. We hope that these revisions will make the logic of the manuscript clearer and make it easier for reviewers and readers to follow.

Minor points:

Page 1 line 24: “Global cancer statistics report that there will be”, this is referring to a 2018 statistic, it should be the past tense “were”.

Our response: We thank the reviewer point out this, and we have modified this error in revised manuscript.

Page 1 line 44: “capacity to flexibly transport their contents to specific sites” what does this mean?

Our response: This means that OMVs can be genetically engineered to carry antigens and some histophagic proteins to target APCs and other tissues or organs. With the guidance of some leader peptides such as ClyA, AIDA-I and OspA (introduced in this paper), the heterologous pathogenic antigens or tumor antigens can be delivered by OMVs. And that, these antigens can be delivered by OMV itself to interact with the endothelial cells, APCs and tumor cells (for example, over-expression of HER2), and thus presented to the specific location. Therefore, we have replaced this sentence by “Coupled with their capacity to flexibly transport their delivered antigens to endothelial cells or antigen presenting cells (APCs), this suggests that OMVs have great potential in vaccine development” in lines 52-53 of revised manuscript.

Page 2 line 7: T-VEC has already been defined in the introduction.

Our response: we have modified this error in revised manuscript.

Page 2 line 69: “Cancer vaccine holds a glove in reducing exposure risk” what is meant by this?

Our response: The cancer vaccine mentioned here is mainly to show that prevention is more important than treatment in its role. This quote uses a metaphor that when a cancer vaccine is used, people who are exposed to risk factors will be as protective as wearing a pair of gloves. Considering the context, we will delete this sentence in the revised manuscript if it will cause misunderstanding.

Page 2 line 51: “prevents the development of cancer in certain high-risk individuals” please give a reference for this.  You should also provide references for the statements in lines 71-74 of the same page.

Our response: We thank the reviewer point out this, and we have added the references, including “Berzofsky, J.A.; Terabe, M.; Trepel, J.B.; Pastan, I.; Stroncek, D.F.; Morris, J.C.; Wood, L.V. Cancer vaccine strategies: translation from mice to human clinical trials. Cancer Immunol Immunother 2018, 67, 1863-1869, doi:10.1007/s00262-017-2084-x; Osipov, A.; Murphy, A.; Zheng, L. From immune checkpoints to vaccines: The past, present and future of cancer immunotherapy. Advances in cancer research 2019, 143, 63-144, doi:10.1016/bs.acr.2019.03.002; Wong, K.K.; Li, W.A.; Mooney, D.J.; Dranoff, G. Advances in Therapeutic Cancer Vaccines. Advances in immunology 2016, 130, 191-249, doi:10.1016/bs.ai.2015.12.001” in line 62 of revised manuscript. And that, we have added the references, including “Berzofsky, J.A.; Terabe, M.; Trepel, J.B.; Pastan, I.; Stroncek, D.F.; Morris, J.C.; Wood, L.V. Cancer vaccine strategies: translation from mice to human clinical trials. Cancer Immunol Immunother 2018, 67, 1863-1869, doi:10.1007/s00262-017-2084-x; Banchereau, J.; Palucka, K. Immunotherapy: Cancer vaccines on the move. Nature Reviews Clinical Oncology 2017, 15, 9; Tran, T.; Blanc, C.; Granier, C.; Saldmann, A.; Tanchot, C.; Tartour, E. Therapeutic cancer vaccine: building the future from lessons of the past. Seminars in immunopathology 2019, 41, 69-85, doi:10.1007/s00281-018-0691-z” in line 89 of revised manuscript.

Page 3 line 123: “OMVs have a proper size”, what is a ‘proper’ size?

Our response: Martin F. Bachmann and Gary T. Jennings et al. mentioned in their review (Bachmann, M.F.; Jennings, G.T. Vaccine delivery: a matter of size, geometry, kinetics and molecular patterns. Nature Reviews Immunology 2010, 10, 787-796) that APCs have evolved to effectively process antigens with dimensions that are similar to pathogens, ranging from 20 to 100 nm. And that, the dimensions of OMVs are commonly ranging from 20 to 200 nm. Therefore, OMVs have a proper size to entry and uptake by APCs. We have modified this sentence and replaced it by “OMVs have proper size (20–200 nm), which is an important factor for their efficient process by APCs, and can present a range of surface antigens in a native conformation, enabling their entry into lymph vessels and uptake by APCs” in lines 134-135 of revised manuscript.

I’d recommend reducing the number of abbreviations in the article – especially where some appear to not be used again.

Our response: We thank the reviewer’s suggestion. And we have carefully checked the use of abbreviations in the full text to ensure simplicity and clarity.

It would be useful to clarify what type of extracellular vesicles gram positive bacteria produce and how these fit in with OMV vaccines. The vesicles themselves weren’t really mentioned until the conclusions and future directions section (with some mention of gram positive bacteria earlier).

Our response: We appreciate the reviewer’ suggestion. We have added the descriptions of extracellular vesicles in the part of introduction, and demonstrated the origin and relationship of the vesicles of Gram-negative bacteria and Gram-positive bacteria. And the detail is as below: Like mammalian cells, Gram-negative and Gram-positive bacteria release nano-sized membrane vesicles into the extracellular environment [12,13]. In recent decades, research into EVs from Gram-negative bacteria increased substantially, but there was little to no EV-related research with Gram-positive bacteria, due to the inference that the thick cell wall of Gram-positive bacteria precluded their existence [14,15]. By contrast, EVs from Gram-negative bacteria originate from the outer membrane and are thus usually referred to as outer membrane vesicles (OMVs) [16]. And OMVs are a lipid-based vesicular nanostructures that contain a variety of porins and can carry heterologous substances to fulfill the adjuvant and delivery function [17]. Coupled with their capacity to flexibly transport their delivered antigens to endothelial cells or antigen presenting cells (APCs), this suggests that OMVs have great potential in vaccine development [18-20]. Although a few studies have reported the antitumor effect of EVs from Gram-positive bacteria, its formation mechanism and biological function have not been fully elucidated. Therefore, in this review, we mainly focus on the OMVs as the EVs from Gram-negative bacteria and analyze the status of cancer vaccine research and the advantages of OMVs as a carrier, focusing on the requirements for designing OMVs as antigen carriers for cancer treatment” in lines 44-58 of revised manuscript. And that, the EVs of Gram-positive bacteria have not been studied thoroughly, and can not be summarized systematically, but many Gram-positive bacteria are used as living bacterial vectors in the development of cancer vaccines and there is one study testing the antitumor effect of EVs from Gram-positive bacteria (Kim, O.Y.; Park, H.T.; Nth, D.; Choi, S.J.; Lee, J.; Kim, J.H.; Lee, S.W.; Gho, Y.S. Bacterial outer membrane vesicles suppress tumor by interferon-γ-mediated antitumor response. Nature Communications 2017, 8, 626). Therefore, we list several Gram-positive live bacterial vectors to provide some novel strategies for design of future cancer vaccines based on EVs of Gram-positive bacteria.

Reviewer 2 Report

The authors provide a summary of the advantages and prospects of outer membrane vesicles of Gram-negative bacteria as antigen-carrier vaccines in cancer vaccine development. There are several points/issues that need to addressed before the manuscript is acceptable for publication. These will be broken down for each section.

Section 1. Introduction

Line 35: Remove extra “of” between words advanced and melanoma.

Section 2. Vaccine in cancer therapy

This section describes the current prophylactic and therapeutic cancer vaccines, but it does not describe what place OMVs take among the total variety of vaccines. Especially since OMVs are adjuvants for cancer vaccines. It would be more logical to review the adjuvants rather than the vaccines.

Section 3. New OMV applications: tumor vaccines

Lines 139, 141, 142, 160, 218, 225-226: Put the genus/species and in vitro in italics.

Line 144: In the section ‘Mechanism of OMVs design as antigen-carrier vaccines’ the authors describe the theoretical possibility of delivering tumor antigens using OMVs but do not give any specific examples.

Line 173: The authors describe in details the mechanism of OMV internalization, but do not describe the subsequent fate of OMVs after uptake and the mechanisms mediating the induction of an immune response. The description of the mechanisms of anti-tumor immune response modulation should be provided. Figure 1 is uninformative, since it also lacks visualization of the main mechanisms responsible for the induction of anti-tumor immune response and does not properly reflect the mechanisms of OMV penetration into host cell described in the text. Also, some elements in the picture are not defined. The Figure should be removed or improved. Detailed description of Figure 1 should also be provided.

Line 208: It is not clear why the authors describe bacterial strains from which it is theoretically possible to obtain OMVs and do not give examples of its actual deriving. Table 1 contains a lot of information about specific strains used for cancer treatment but there is no data on its therapeutic efficacy.

Line 260: typo (excessive full stop).

Line 276: Appropriate reference should be provided.

Author Response

Reviewer 2

The authors provide a summary of the advantages and prospects of outer membrane vesicles of Gram-negative bacteria as antigen-carrier vaccines in cancer vaccine development. There are several points/issues that need to addressed before the manuscript is acceptable for publication. These will be broken down for each section.

Our response: We appreciate the reviewer’s suggestions, and with the help of reviewers, the quality of our manuscripts has been greatly improved.

Section 1. Introduction

Line 35: Remove extra “of” between words advanced and melanoma.

Our response: We have deleted this word in revised manuscript.

Section 2. Vaccine in cancer therapy

This section describes the current prophylactic and therapeutic cancer vaccines, but it does not describe what place OMVs take among the total variety of vaccines. Especially since OMVs are adjuvants for cancer vaccines. It would be more logical to review the adjuvants rather than the vaccines.

Our response: In this part, we summarize the advantages and disadvantages of prophylactic and therapeutic vaccines, and think that therapeutic vaccines should be the development direction and option of cancer treatment. Since we have not yet specified the characteristics of OMVs, we did not consider introducing OMV into this section at first. But in order for reviewers and readers to follow our content better, we have added these sentences of that “Owing to its ability of immune regulation and presenting multiple antigens to host cells, OMVs can be a new option for use as an effective therapeutic vaccine by delivering tumor antigens or small molecule drugs to APCs cells or even targeted cancer cells through genetic engineering” in lines 77-80 of this section to illustrate the relationship between OMV and these vaccines.

Further, because OMVs laden with PAMPs, it has a certain adjuvant function to stimulate the host's immune response. But this is not all the function of OMVs. More importantly, OMVs can present multiple antigens and improve the targeting efficiency of antigens by genetic engineering of host bacteria. This is the key to the design of OMV as a new generation of cancer vaccine. Therefore, we can not simply describe OMVs as an adjuvant for cancer vaccine. Perhaps inappropriate language organization in previous versions would lead to misunderstanding between reviewers and readers. We have made a lot of modifications to the manuscript. Firstly, we have added some sentences of that “What we are concerned about here is how to improve the protective effect of the vaccine stimulating to the host and the delivery efficiency of the vaccine vector. And we believe that OMVs as a tumor antigen carrier can provide us with a novel strategy to design cancer vaccines” to emphasize the possible function of OMV as a tumor antigen presenting vector in lines 89-92 of revised manuscript. Secondly, we have added the new Table 1 and reorganized the part of “3.1 Roles of OMVs as vaccines” to demonstrate the function of OMVs as vaccines. We hope that this revised version will satisfy the reviewers.

Section 3. New OMV applications: tumor vaccines

Lines 139, 141, 142, 160, 218, 225-226: Put the genus/species and in vitro in italics.

Our response: We thank the reviewer for point out these errors, and we have modified all errors in revised manuscript.

Line 144: In the section ‘Mechanism of OMVs design as antigen-carrier vaccines’ the authors describe the theoretical possibility of delivering tumor antigens using OMVs but do not give any specific examples.

Our response: We appreciate the reviewer’s suggestion. In this part, we mainly focus on introducing the mechanisms. Some specific examples have been shown in the references. And because of the space limitation of the paper, we did not add specific examples at the beginning. But as the reviewer said, adding specific examples can help readers follow our content better, we have added the specific examples at lines 201-207 to demonstrate these mechanisms of OMV entry into cells. And the detail text is as below: “1. Since Helicobacter pylori OMVs are known to transport vacuolating toxin VacA, an important cytotoxic virulence factor during infection, they have been shown to facilitate their cargo delivery via clathrin mediated endocytosis [79]. 2. Sharpe and colleagues identified this process in nontypeable Haemophilus influenzae and showed that OMVs colocalized with the endocytosis protein caveolin, indicating that internalization is mediated by caveolae, which are cholesterol-rich lipid raft domains [81]”. We hope that these revisions will satisfy the reviewer.

Line 173: The authors describe in details the mechanism of OMV internalization, but do not describe the subsequent fate of OMVs after uptake and the mechanisms mediating the induction of an immune response. The description of the mechanisms of anti-tumor immune response modulation should be provided. Figure 1 is uninformative, since it also lacks visualization of the main mechanisms responsible for the induction of anti-tumor immune response and does not properly reflect the mechanisms of OMV penetration into host cell described in the text. Also, some elements in the picture are not defined. The Figure should be removed or improved. Detailed description of Figure 1 should also be provided.

Line 208: It is not clear why the authors describe bacterial strains from which it is theoretically possible to obtain OMVs and do not give examples of its actual deriving. Table 1 contains a lot of information about specific strains used for cancer treatment but there is no data on its therapeutic efficacy.

Our response: In Part 3.4 Bacterial OMVs as cancer vaccines, we list some cases of using E. coli as parent bacteria to modify its OMV to design cancer vaccine, which can be regarded as the extension of the live bacterial vector used in the development of cancer vaccine. Table 1 mainly lists living bacterial vectors that can be used for cancer antigen presentation, and it can provide reference for future design of OMV-based tumor antigen presenting vaccine. And that, at the present stage, many studies only directly use OMVs derived from conventional engineering of E. coli. Genetic engineering of OMVs presents great potential to artificially modify different intracellular bacteria such as Salmonella, Listeria, to maximize the ability of OMVs to stimulate immune responses, thereby designing an ideal cancer vaccine. And we have mentioned this in the part of conclusion. However, there is no research on the development and design of cancer vaccines based on Salmonella or Listeria OMVs. This is the original intention of our review, and we hope that more researchers will be involved in the development of OMV-based cancer vaccines. Therefore, we can not list specific example of OMV in this table. As a supplement, we add the example of direct use of EVs from Gram-positive bacteria in anti-tumor research in lines 404-407 of revised manuscript. Moreover, we have reorganized the first paragraph of the part 3.4-Bacterial OMVs as cancer vaccines to demonstrate the relationship between OMV-based cancer vaccine and live bacterial vector.

For the second question, we have added the data of clinical trials and therapeutic efficacy of various live bacterial vectors, and the detail revisions can be seen in revised Table 2. We hope these modifications will satisfy the reviewer.

Line 260: typo (excessive full stop).

Our response: We have modified this error in line 318 of revised manuscript.

Line 276: Appropriate reference should be provided.

Our response: We thank the reviewer’s suggestion, and we have added the reference at line 333 to demonstrate the issue that OMV could use EPR effect to deliver drugs and release into the vicinity of tumor cells (Ernsting, M.J.; Murakami, M.; Roy, A.; Li, S.D. Factors controlling the pharmacokinetics, biodistribution and intratumoral penetration of nanoparticles. Journal of controlled release : official journal of the Controlled Release Society 2013, 172, 782-794, doi:10.1016/j.jconrel.2013.09.013; Tahmasbi Rad, A.; Chen, C.W.; Aresh, W.; Xia, Y.; Lai, P.S.; Nieh, M.P. Combinational Effects of Active Targeting, Shape, and Enhanced Permeability and Retention for Cancer Theranostic Nanocarriers. ACS applied materials & interfaces 2019, 11, 10505-10519, doi:10.1021/acsami.8b21609). And this reference showed that the efficiency of targeting tumor cells strongly depends on the physicochemical characteristics of nanoparticles, including both size and shape. And it has been established that spherical particles with a diameter range of 20–100 nm result in optimal tumor accumulation because of the EPR effect.

Round 2

Reviewer 2 Report

The authors answer to all my comments.

Author Response

Thank you very much for your positive comments to our reply of round 1. We have invited experts of Editage, which is a professional company for paper edition to edit this manuscript (www.editage.cn). The expert has read through this manuscript and edited the grammar and typing errors. We hope this revised manuscript looks concise and clear.